# Discrimination of Genetically Very Close Accessions of Sweet Orange (*Citrus sinensis* L. Osbeck) by Laser-Induced Breakdown Spectroscopy (LIBS)

**DOI:** 10.3390/molecules26113092

**Published:** 2021-05-21

**Authors:** Aida B. Magalhães, Giorgio S. Senesi, Anielle Ranulfi, Thiago Massaiti, Bruno S. Marangoni, Marina Nery da Silva, Paulino R. Villas Boas, Ednaldo Ferreira, Valdenice M. Novelli, Mariângela Cristofani-Yaly, Débora M. B. P. Milori

**Affiliations:** 1Embrapa Instrumentation, São Carlos 13560-970, SP, Brazil; aida.magalhaes@agrorobotica.com.br (A.B.M.); aniranulfi@gmail.com (A.R.); thiagomassaiti_k_k@hotmail.com (T.M.); marina.nery.s@gmail.com (M.N.d.S.); paulino.villas-boas@embrapa.br (P.R.V.B.); ednaldo.ferreira@embrapa.br (E.F.); debora.milori@embrapa.br (D.M.B.P.M.); 2Agrorobótica, São Carlos 13571-512, SP, Brazil; 3CNR, Istituto per la Scienza e Tecnologia dei Plasmi (ISTP), Sede di Bari, 70126 Bari, Italy; 4São Carlos Institute of Physics, University of São Paulo, São Carlos 13566-590, SP, Brazil; 5Physics Institute, Federal University of Mato Grosso do Sul, Campo Grande 79070-900, MS, Brazil; bruno.marangoni@gmail.com; 6Instituto Agronômico, Centro de Citricultura Sylvio Moreira (IAC-CCSM), Cordeirópolis 13490-000, SP, Brazil; valdenice@ccsm.br (V.M.N.); mariangela@ccsm.br (M.C.-Y.)

**Keywords:** sweet orange accessions, laser-induced breakdown spectroscopy, elemental concentration, classification via regression (CVR), partial least square regression (PLSR)

## Abstract

The correct recognition of sweet orange (*Citrus sinensis* L. Osbeck) variety accessions at the nursery stage of growth is a challenge for the productive sector as they do not show any difference in phenotype traits. Furthermore, there is no DNA marker able to distinguish orange accessions within a variety due to their narrow genetic trace. As different combinations of canopy and rootstock affect the uptake of elements from soil, each accession features a typical elemental concentration in the leaves. Thus, the main aim of this work was to analyze two sets of ten different accessions of very close genetic characters of three varieties of fresh citrus leaves at the nursery stage of growth by measuring the differences in elemental concentration by laser-induced breakdown spectroscopy (LIBS). The accessions were discriminated by both principal component analysis (PCA) and a classifier based on the combination of classification via regression (CVR) and partial least square regression (PLSR) models, which used the elemental concentrations measured by LIBS as input data. A correct classification of 95.1% and 80.96% was achieved, respectively, for set 1 and set 2. These results showed that LIBS is a valuable technique to discriminate among citrus accessions, which can be applied in the productive sector as an excellent cost–benefit tool in citrus breeding programs.

## 1. Introduction

Accessions of sweet orange varieties result in different combinations of canopy and rootstock and show a wide diversity of morphological/phenotype traits, such as size and shape of the canopy, color and size of the fruit, and ripening time and quantity of seeds per fruit. These features can be identified by visual inspection of adult plants and are widely used to discriminate accessions in the period of fruit production, whereas in other phenological stages the plants do not show any difference in these traits and are not distinguishable. Although sweet orange accessions exhibit a substantial diversity of morphological traits, the genetic differences at the molecular level are very subtle [1] as they feature a narrow genetic basis due to somatic mutations and perpetuation by vegetative propagation [2,3]. Thus, none of several DNA markers available for the identification of sweet orange is capable of distinguishing among accessions due to their low polymorphism.

Due to this, the producers face a problem in identifying correctly orange accessions in the commercial genotypes, which may lead to erroneous cultivations with several consequences for the sector. According to Radman and Oliveira [4], different rootstocks may influence the biological processes occurring in orange plants, including the uptake of nutrients and water from soil, transpiration, chemical composition of leaves, etc. Thus, different combinations of rootstock and canopy may result in different chemical and biological properties of the plants, including the concentration of elements in leaves.

As orange is an economically very important crop in Brazil and worldwide, the development of tools that can help to characterize accessions in the early stages of plant growth is very relevant and extremely desirable. To assist citrus breeding programs and advance the research on the discrimination of orange variety accessions in Brazil, the Optic and Photonic Laboratory of Embrapa Instrumentation and the Agronomic Institute of Campinas (IAC) joined efforts to establish and validate a rapid and efficient method for the identification and discrimination of orange accessions. To this purpose, laser-induced fluorescence spectroscopy (LIFS) showed to be a promising suitable certification tool able to discriminate orange accessions [5,6,7].

Differently from LIFS, which is a molecular analytical technique, laser-induced breakdown spectroscopy (LIBS), is able to perform elemental analysis by measuring the spectral emissions of the analyte in the plasma rapidly, in real-time and with no or very limited sample preparation, avoiding the use of chemical reagents and providing high sensitivity and reliable data [8,9,10,11,12]. Furthermore, the development of in situ LIBS settings allows portability and large-scale operations and reduces the risk of sample loss or contamination [13]. In the last decade, a few reviews [14,15,16] have surveyed comprehensively the applications of LIBS combined with modern powerful chemometric methods to agricultural materials and products, including plants, (leaves, roots, fruits, and grains). In particular, Santos et al. [14] have reviewed the qualitative and quantitative LIBS analysis of plant materials such as fresh and dried leaves, fruits, and roots applied to plant nutrition diagnosis and elemental mapping. Peng et al. [15] have introduced and reviewed some signal enhancement methods to improve LIBS performance and calibration methods for quantitative analysis of plants, agricultural products, and food. Senesi et al. [16] have reviewed the recent progress and challenging applications of multi-elemental LIBS to the analysis of plants, agricultural products, and food with particular attention to fruits and grains. The potential of recent advanced LIBS approaches, including double-pulse (DP)-LIBS, femtosecond (fs)-LIBS, micro-LIBS, nanoparticle-enhanced (NE) LIBS, and 3D elemental imaging have been also discussed.

The objective of this work was to possibly identify and highlight very subtle variations in the elemental composition of sweet orange accessions using the spectral profile measured by LIBS combined with chemometric methods. To this purpose, LIBS elemental data were acquired on two sets of leaves collected from ten sweet orange accessions of three different varieties at the nursery stage grown in greenhouses and submitted to the same type of fertilization and irrigation. Then, LIBS data were processed by principal component analysis (PCA) and a combined classification via regression (CVR) and partial least square regression (PLSR) analysis, with the aim of obtaining a classifier able to yield the corresponding confusion matrices and so discriminate the accessions on dependence on the combinations of canopy and rootstock.

## 2. Results and Discussion

### 2.1. The LIBS Spectrum

A typical orange leaf LIBS spectrum is shown in Figure 1, where the entire spectral range is split into two consecutive wavelength ranges and the relevant elemental lines identified. In particular, twenty-three atomic and ionic emission lines (Table 1) were identified and assigned to the corresponding elements using the NIST database [17]. The elements Ca, Mg, Na, and K are considered plant macronutrients, Zn and Fe micronutrients, and Ti at low concentration beneficial, whereas Al is considered a toxic element and Si as an element not relevant to plant nutrition.

The LIBS spectra of leaves were employed to discriminate among genetically close orange accessions based on the amount of each element. To this purpose, LIBS spectra were submitted to the paired Student’s *t*-test at 95% confidence level by comparing two by two the accessions within each set. The p-values of the selected lines were all below 0.05, which supported the existence of statistically significant differences among the accessions considered.

### 2.2. Principal Component Analysis and Classifier Model Based on Some Selected LIBS Elemental Lines

The elemental emission lines in Table 1 were analyzed by PCA separately for each variety in each set using the areas under the peaks of the elements selected as input variables. Figure 2, Figure 3, Figure 4 and Figure 5 and Figure 7 show the 2-dimensional score plots, the scree plots, and the loading plots for the two first PCs, PC1xPC2, which retain most of the variance. In all cases the total variance retained in the first two PCs did not increase by increasing the number of input attributes.

Figure 2 shows the scree, score, and loading plots of the four accessions of the Common variety of set 1. The scree plot shows that the variance explained by the two first PC retains most of the total variance, i.e., 70.89%, needed for the differentiation among the four accessions. The 2D score plot shows that data could be grouped into three somehow overlapping main clusters with the accessions Pera Ipigua and Pera Mahle superimposed. The loading plot indicates that the areas of Mg and Ca peaks correlate positively with PC1, i.e., as the elemental concentrations increase PC1 also increases, whereas an opposite behavior is shown for PC2, i.e., as their concentrations increase PC2 decreases. The peak areas of Zn I, Si I, Al I, Ti II, K I, and Na I appear to correlate with and contribute positively to both PC1 and PC2, whereas the Fe I line at 302.10 nm shows a weak positive contribution to both PC1 and PC2.

The scree, score, and loading plots of the two accessions of the Navel variety of set 1 are shown in Figure 3. Additionally, in this case most variance, i.e., 75.73%, is explained by the first two principal components (Figure 3a). The score plot shows a partial overlap of clusters, although the centers of the two clusters tend to separate (Figure 3b). The loading plot (Figure 3c) of the accessions of this variety is very similar to that of the Common variety of set 1 described above.

The scree plot (Figure 4a) of the four accessions of the Pigmented variety of set 1 show that PC1 and PC2 contribute 67.72% of the total variance and the score plot features no apparent separation of the four clusters (Figure 4b). In this case, only the peak areas (Figure 4c) of Ca II and Ca I show an opposite correlation with the two PCs, i.e., positive with PC1 and negative with PC2. Differently from the other two varieties of set 1, the Zn, Mg, Si, Fe, Al, Ti, Na, and K peak areas show a positive correlation for both PC1 and PC2.

Similar to the other varieties described above, the scree plot of the six accessions of the Common variety of Set 2 shows that the first two PCs retain most of the total variance, i.e., 64% (Figure 5a) whereas the score plot does not show any apparent separation (Figure 5b). The loading plot (Figure 5c) shows that the peak areas of Ca, K, and Mg contribute positively to PC1 and negatively to PC2, whereas the remaining elements, i.e., Zn, Si, Fe, Al, Ti, and Na appear all to contribute positively especially to PC2. To achieve a more detailed analysis distinct PCA plots were constructed for each pair of accessions of this variety (Figure 6). The main results are summarized below: (i) the cluster mass center of Pera Bianchi accession shows a tendency to separate from those of Seleta do Rio, Pera Abril IAC 148, and Valencia B accessions; (ii) the Pera Roberto Gullo accession cluster tends to separate from almost all other accessions, except the Pera Bianchi one, which suggests that this accession possesses characteristics different from the other ones; (iii) the Valencia B accession cluster is slightly separated only from Pera April IAC 148, thus it seems to be very similar to the others.

The scree plot of the three accessions of Low Acidity variety of set 2 (Figure 7a) shows that the first two PCs explain 57.92% of the total variance. The score plot (Figure 7b) shows that the Piralima IAC 2 cluster is superimposed to those of the other two varieties, Lima IAC 9 and Piralima IAC 11, which appear slightly separated from each other. The loading plot (Figure 7c) shows that the peak areas of Mg, Ca, and K have a positive correlation with PC1 and a negative one with PC2. The elements Si, Al, Ti, and Na contribute positively to both PC1 and PC2, whereas Zn and Fe have almost no influence on both PCs. Finally, PCA could not be performed for the Pigmented variety of set 2, which consists of only one accession.

As PCA applied to selected peak areas allows to discriminate only among some citrus accessions, the discrimination was attempted by constructing an induced classifier based on a multivariate classification model obtained by combining CVR and PLSR procedures applied to all accessions within each set of leaves examined, using as input variables the same twenty-three peak areas used previously for PCA. Data in Table 2 show that an appreciable accuracy is achieved by the model, i.e., between 90% and 100% and between 86.5% and 65% of correctly classified instances for set 1 and set 2, respectively. These results suggest that LIBS combined with a machine-learning method is a very promising tool for discrimination among citrus accessions on dependence on each combination of canopy and rootstock which determine plant nutritional availability.

### 2.3. Classifier Model Based on the Entire Broadband LIBS Spectra

A combined CVR and PLSR approach was then applied to the entire broadband LIBS spectra of all citrus accessions of set 1 and set 2 in the attempt of achieving a better classification model. The confusion matrices generated for set 1 and set 2 samples, respectively, Table 3 and Table 4 indicate that a variable accuracy could be achieved by the LIBS data based model in discriminating between sweet orange accessions. In particular, almost all accessions of set 1 (Table 3) are correctly classified (100%). Although, some misclassifications/confusion occur for one Common and two Pigmented variety accessions, which suggest that the genetic traits of these accessions are so close that their discrimination is not completely efficient. No accession of set 2 is classified correctly (Table 4), with a variable level of confusion for various accessions in the same variety and/or other varieties. The higher misclassifications achieved among set 2 accessions with respect to those of set 1 may be ascribed to the farther genetic traits of the former ones with respect to the latter ones. The performance of the classification model, i.e., sensitivity and specificity, yield average values of 98.5% and 99.83 for set 1 and 70% and 96.22% for set 2.

A better classification was attempted by applying the procedure to LIBS data of the accessions of each variety separately. The confusion matrix achieved for the Common variety of set 1 shows in all cases a success rate of 100% instances classified correctly (Table 5), whereas for set 2 of this variety the instances classified correctly are between 70% and 95% (Table 6), with the major confusion found for Pera Abril IAC 148 (A) and Seleta do Rio (D). The model shows an average sensitivity and specificity of both 100% for this variety in set 1 and 82.5% and 97.08% in set 2.

The confusion matrix obtained for the Pigmented variety accessions of set 1 (Table 7) shows 100% correctly classified instances for three of them, with only one low confusion found for Red Pulp of Mombuca with Sanguinea accessions. An average sensitivity and specificity of 81.25% and 95.31% are achieved for this variety. As the Pigmented variety of set 2 included only one accession, no classifier could be applied.

The confusion matrix achieved for the two accessions of the Navel variety of set 1 (Table 8) shows 100% of instances classified correctly with 100% at both sensitivity and specificity. These two accessions do not show any incorrect instance even in the model generated when all other accessions are considered, thus confirming that their leaf elemental composition are very different from each other and from those of the other leaf samples.

Finally, the confusion matrix generated for the accessions of the Low Acidity variety (Table 9) shows a good success rate with an average 90% correctly classified instances. The average sensitivity and specificity achieved for this variety are both 100%.

## 3. Materials and Methods

### 3.1. Leaf Samples

Two sets of leaf samples, each including 10 accessions of three different varieties of sweet orange (*C. sinensis L.* Osbeck) (Table 10) were analyzed. The first set was collected in winter and the second one in summer, both from nursery plants grown in a greenhouse in the Centro de Citricultura Sylvio Moreira (CCSM) at the Citrus Germplasm Bank of the Agronomic Institute of Campinas (IAC). All plants were grafted on C. *limonia* and maintained in 60-L citrus pots containing a commercial substrate in greenhouse conditions. All growth parameters, including soil management, irrigation, fertilization, temperature, and watering, were adequate and favorable for a correct plant growing. All plants were about 3 years old, i.e., between the seedling and adult phases. At this stage, the accessions did not show any difference in phenotype traits, which did not allow any differentiation either by visual inspection or by any molecular marker.

Ten plants were selected for each accession and three leaves were collected from each plant for a total of 30 leaf samples per accession. Leaves were cleaned with cotton wool to remove any soil and dirt that could interfere with LIBS analysis. All measurements were performed on the fresh leaf with minimal preparation. The central vein of the fresh leaf was removed to eliminate any surface roughness and make it easier leaf positioning in the sample holder in the ablation chamber. The LIBS analysis was performed within 24 h after collection. During this time, the leaves were maintained in bags and refrigerated at 4 °C to avoid degradation and drying [7].

### 3.2. Laser-Induced Breakdown Spectroscopy Equipment and Spectra Acquisition

The LIBS spectra were acquired using a commercial equipment consisting of an LIBS2500 spectrometer (Ocean Optics, Dunedin, FA, USA) provided with a Q-switched 1064 nm Nd:YAG laser (Quantel, Big Sky Laser Ultra50), operating at a single wavelength of 1064 nm with a pulse energy of 50 mJ, a repetition rate of 10 Hz and a laser pulse of 8 ns, and an initial delay time fixed at 2.5 µm. The laser fluence value was 1200 Jcm^−2^ and the distance between the focusing lens and the sample was 75 mm. All measurements were performed in an ablation chamber in the air at atmospheric pressure. The detection system consisted of a set of seven spectrometers operating in the spectral ranges, respectively, of 188.84–292.61 nm, 292.61–384.44 nm, 384.44–507.16 nm, 507.16–618.30 nm, 618.30–716.20 nm, 716.20–800.25 nm, and 800.25–966.22 nm, with an optical resolution of 0.1 nm and equipped with a charge-coupled device (CCD) array.

The high-energy laser pulses are focused on the sample to generate the plasma and excite thermally the analyte to higher energy levels. As the plasma cools down, the atoms decay to lower energy levels emitting electromagnetic radiation that is collected by a spectrometer that resolves the atomic emissions in wavelength. A typical LIBS spectrum features several narrow spectral lines at specific wavelengths characteristic of the elements present in the sample, the intensity of which is proportional to the element concentration.

Three shots were applied for each measurement, the first one used to clean the sample surface and the other two ones for its ablation. For each leaf sample, ten measurements were performed and averages were calculated to minimize uncertainties due to the biological heterogeneity of samples. Accessions from the same variety were not measured in sequence to avoid interference [7]. The software OOIBase32 developed by Ocean Optics was used for data acquisition.

### 3.3. Treatment of Spectra and Classification Model

The qualitative elemental analysis was performed by using the characteristic elemental peaks selected by applying the paired Student’s *t*-test with 5% of significance to each point of the spectrum by comparing couples of investigated accessions. P-values lower than 0.05 allowed finding the regions of the spectrum showing statistically significant differences between the accessions, with a smaller p-value indicating a smaller difference. Thus, the emission peaks of the elements that mostly differ among the accession could be identified. Then, the baseline of each emission peak was subtracted by the slope of the line defined by two points at each limit boundary of the peak, which was then normalized by the area, A2, calculated under the line according to Marangoni et al. [18]. Finally, the area of the peak, A1, was calculated by fitting a Lorentzian shape over the peak profile. In order to select the fit with the lowest mean error, the command-line Peak Fit program implemented in a routinary Matlab software was used to perform multiple preliminary trial fits with slightly different starting values. This procedure permitted to choose the best fit by returning a FitResults vector with several fitting parameters among which the Mean Fit Error, i.e., the percent RMS difference between the data and the model in the selected segment of that data. Among the several fits evaluated, including the Gaussian and Lorentzian fits, the best one for the transition LIBS spectral lines was the Lorentzian fit.

First, the PCA approach was used to evaluate the potential of LIBS data in discriminating among sweet orange accessions and construct a classifier able to correctly identify each accession. The calculated areas, A1, for each selected element peak from all samples, i.e., 23 peaks per sample, were used as input variables for PCA, which was performed using the software Origin 9.0. PCA is a multivariate unsupervised statistical method able to project multivariate data and describe relevant trends in the analyzed data set by reducing the dimensionality of the original space of dataset without affecting the relations between the samples [19,20]. The reduction of the number of variables is achieved by making a linear combination of original variables, which yields the so-called principal components (PC) that are decorrelated with each other, i.e., the information contained in one is not present in the other and are defined so to describe the direction of maximum variance of the original data. The PCs are orthogonal and oriented one by one to describe the maximum remaining variance. Once redundancy is removed, almost all information contained in the original data can be described by only a few major PCs. In this work, the following PCA information was considered (i) the scree-plot, to estimate the variance of data according to each PC; (ii) the score plot, to visualize the projection of the sample on each PC; and (iii) the loading plot, to evaluate the influence of each variable, i.e., elemental peak areas, on each PC, i.e., the relationship between the original variables and the subspace dimension [21]. Thus, PCA allows emphasizing and interpreting all relevant differences among samples.

Thus, the discrimination among citrus accessions was performed using two classifiers, the one based on 23 selected elemental peak areas, A1, evaluated previously as input variables for PCA, the other employing the whole LIBS spectra. The broadband spectra were previously pre-treated to highlight the relative spectral variations, reduce LIBS signal uncertainties resulting from fluctuations of laser energy and matrix effects, and enable spectra comparison. In particular, the data acquired on each leaf were normalized by the area of the whole spectrum calculated individually for each spectrometer range, i.e., each intensity point within each spectrometer was divided by the area calculated under the curve of the same range. After normalizing the spectra of each measurement, the average of 10 shots per leaf was obtained. All calculations were performed using the routinary Matlab software 2012. To achieve the differentiation between classes, 13,560 spectral points in the average broadband spectrum of each sample were used as input variables in the classifier model. The two data sets consisting of either the 23 selected elemental peak areas or the whole LIBS emission spectra were exported to the open-source software Weka to yield the classifiers obtained by combining CVR and the regression function via PLSR [7,22,23,24].

The final classifier was thus based on the different elemental concentrations in the leaves of different accessions. In both cases, 10 executions of 10-fold stratified cross-validation were performed to verify the classification model. This procedure allowed to separate the data sets into 10 folders, with 9 used for training the classifier and one for testing it. Then, the folder that was used as the test was put back into the dataset and another folder was chosen randomly to carry out the same procedure. The iteration process was repeated until all groups were tested. The randomization within the training and test groups ensured that the accuracy obtained was not based on a particular partition of training and test sets. The output of the induced classifiers provided a confusion matrix, in which the numbers in the rows indicated the actual class, those in the columns the results of the classifier, and the values in the diagonal the correctness of the model. All values were expressed in percentage. To optimize the classifier success rate, the number of components was chosen by analyzing the highest success rate and the lowest root mean squared error (RMSE) as a function of the number of components with significant statistical differences. For all tests, ten components were used.

The performance of the CVR associated with the PLSR classifier was calculated using two parameters, sensitivity, and specificity, which are described in terms of True Positive (*TP*), True Negative (*TN*), False Positive (*FP*), and False Negative (*FN*) according to Equations (1) and (2) below [7,25]:(1)$$Sensitivity=\frac{TP}{TP+FN}$$
(2)$$Specificity=\frac{TN}{TN+FP}$$

## 4. Conclusions

The LIBS technique combined with chemometric methods shows a great potential to discriminate between different and genetically very close accessions of various sweet orange varieties based on the relevant elemental composition of their leaves at the nursery stage that are specific for all combinations of canopy and rootstock. The classification models applied to the elemental composition data extracted from LIBS spectra allow to discriminate the various accessions at various levels of accuracy for each variety examined.

Differently from DNA markers and visual inspection, which can be applied successfully only to adult plants bearing fruits, LIBS has the advantage of being applied to plants at the nursery stage, thus being able to discriminate earlier among the accessions by measuring the relevant elemental composition differences in the leaves. An additional advantage of LIBS is that, when embedded in an appropriate vehicle, the apparatus is transportable, so allowing large-scale and real-time on-site monitoring for early identification of orange accessions in breeding programs.

In conclusion, LIBS represents a novel promising and economically viable spectroscopic tool for the early identification and discrimination at the nursery stage of sweet orange variety accessions featuring very small genetic variability and indistinguishable to molecular markers which is of particular importance for the productive sector, genetic certification, and germplasm core collection selection.

## Figures and Tables

**Figure 1 molecules-26-03092-f001:**
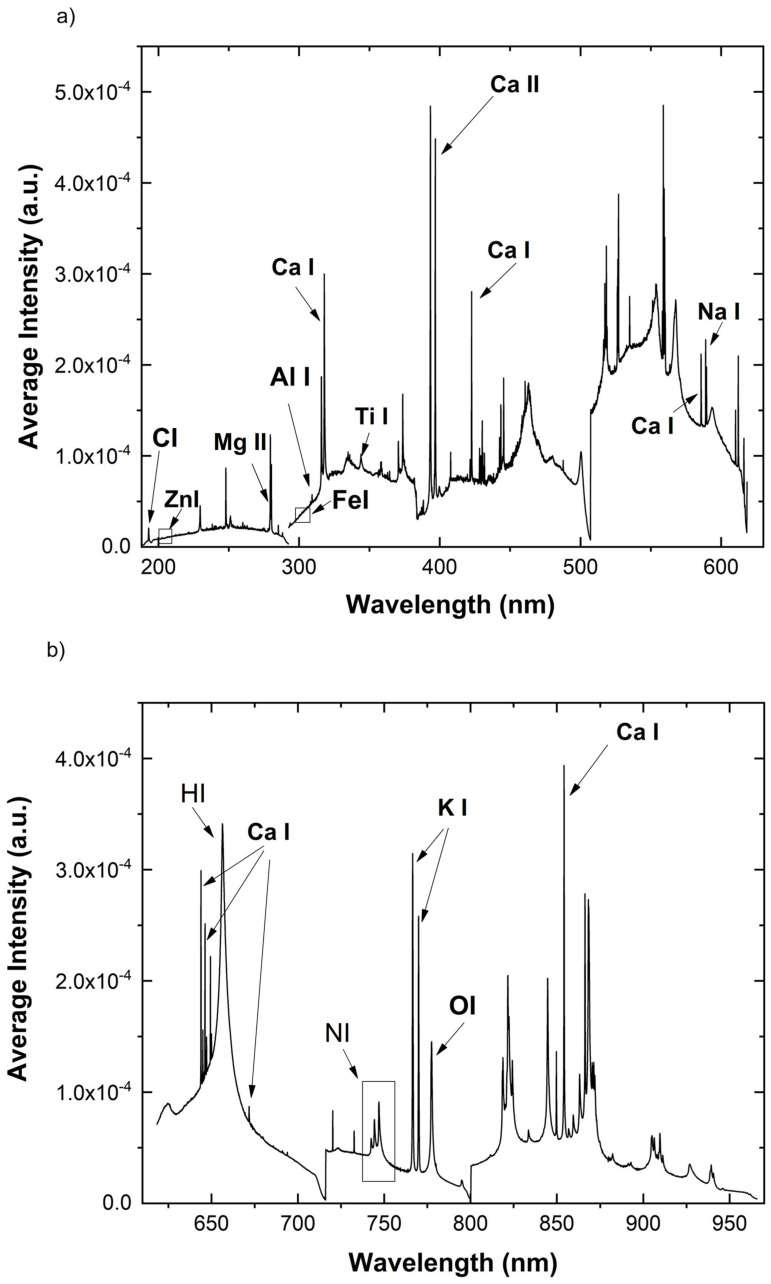
A typical LIBS spectrum of a sweet orange leaf in the spectral ranges 188.84−618.30 nm (**a**) and 618.30 nm−966.22 nm (**b**).

**Figure 2 molecules-26-03092-f002:**
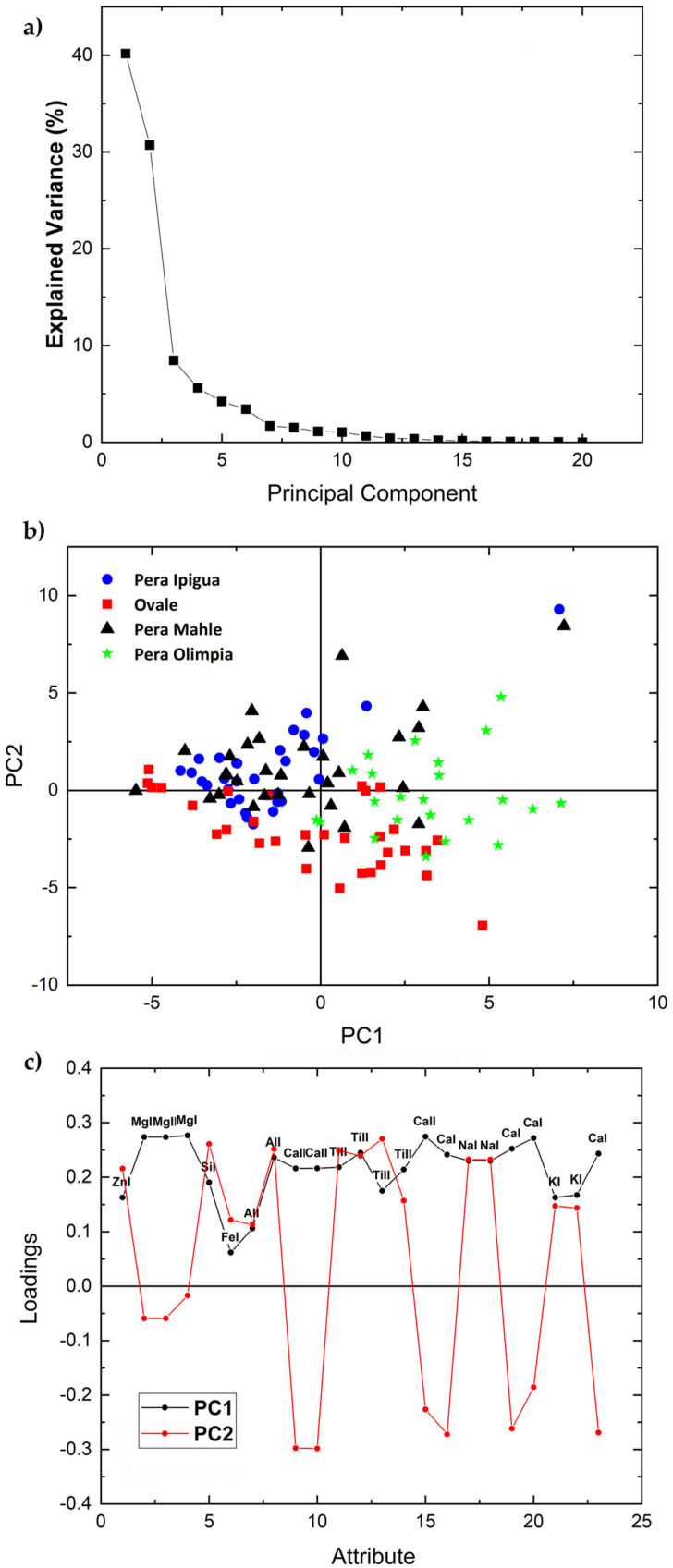
Scree plot (**a**), PCA 2-dimensional score plot of PC2 versus PC1 (**b**), and loading plot showing how each variable correlates to each PC (**c**) of the four accessions of the Common variety of set 1.

**Figure 3 molecules-26-03092-f003:**
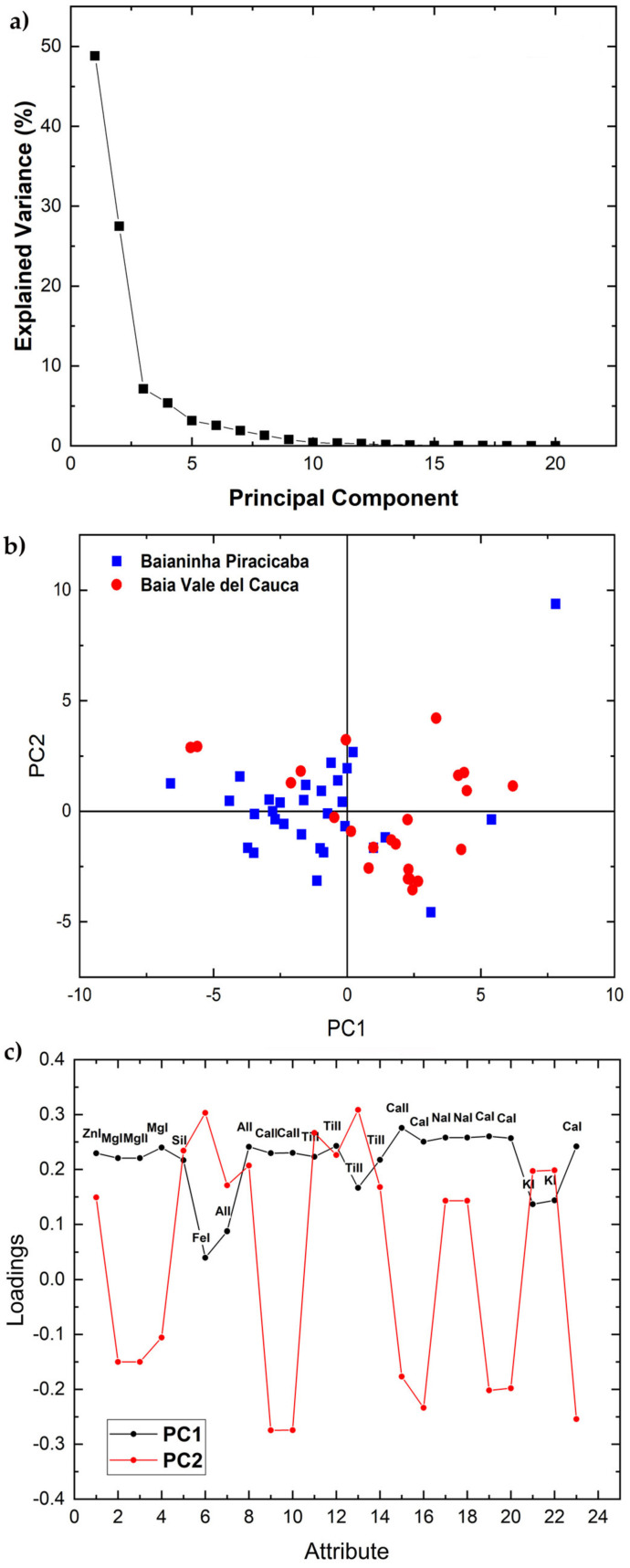
Scree plot (**a**), PCA 2-dimensional score plot of PC2 versus PC1 (**b**), and loading plot showing how each variable correlates to each PC (**c**) for the two accessions of the Navel Variety of set 1.

**Figure 4 molecules-26-03092-f004:**
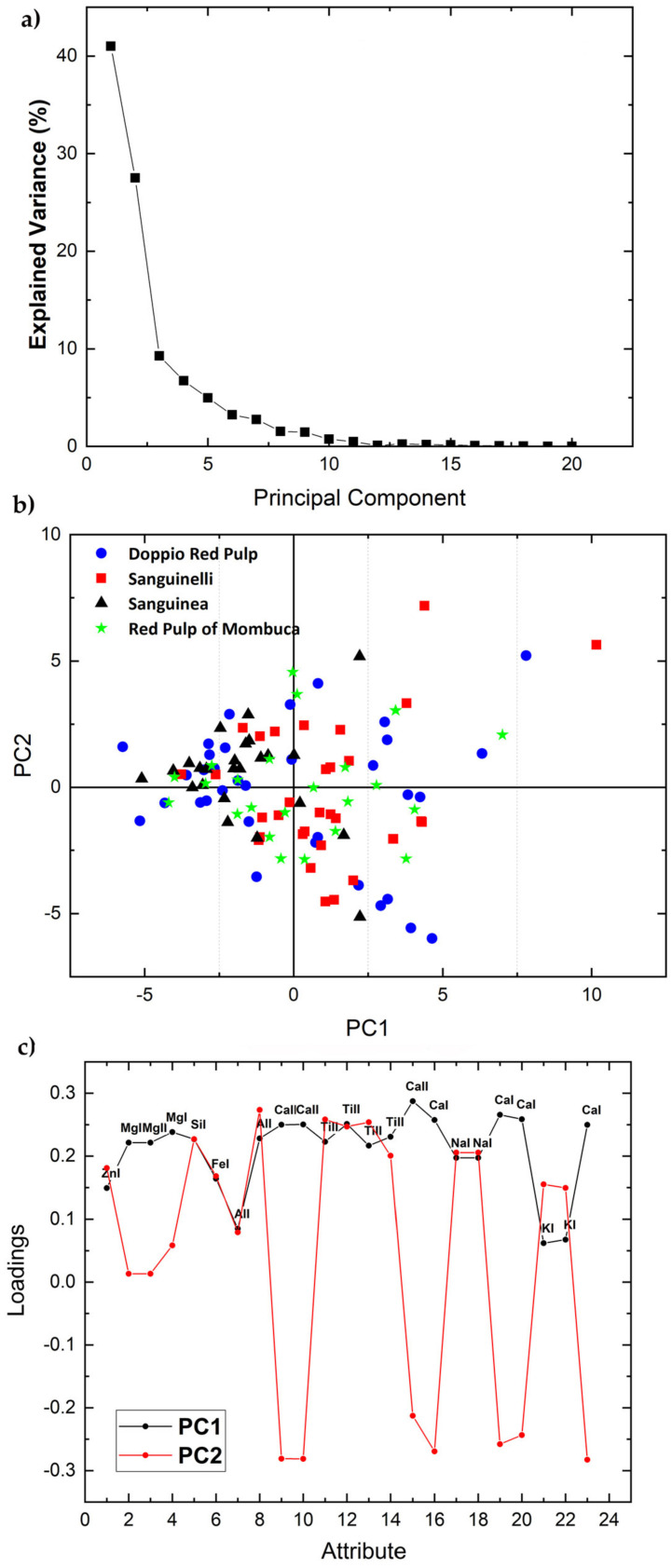
Scree plot (**a**), PCA 2-dimensional score plot of PC2 versus PC1 (**b**), and loading plot showing how each variable correlates to each PC (**c**) for the four accessions of the Pigmented Variety of set 1.

**Figure 5 molecules-26-03092-f005:**
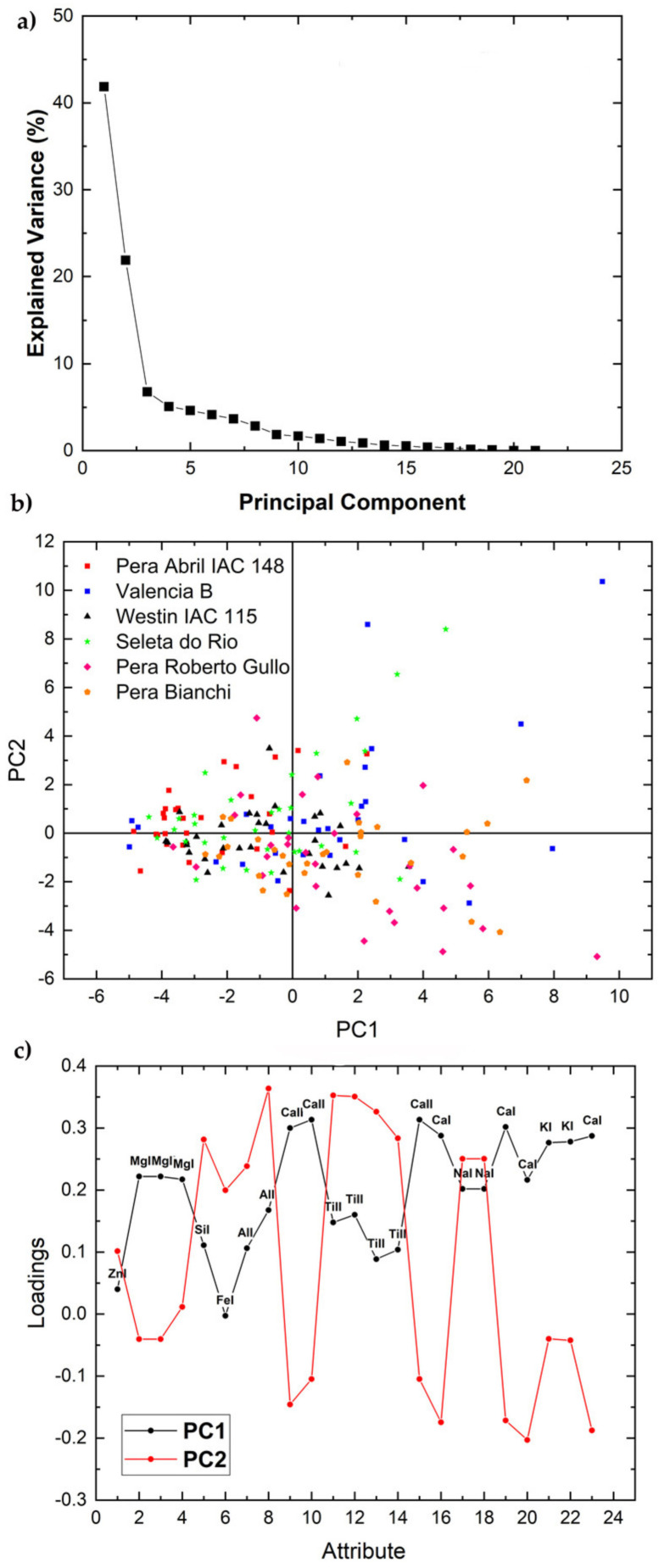
Scree plot (**a**), PCA 2-dimensional score plot of PC2 versus PC1 (**b**), and loading plot showing how each variable correlates to each PC (**c**) for the six accessions of the Common variety of set 2.

**Figure 6 molecules-26-03092-f006:**
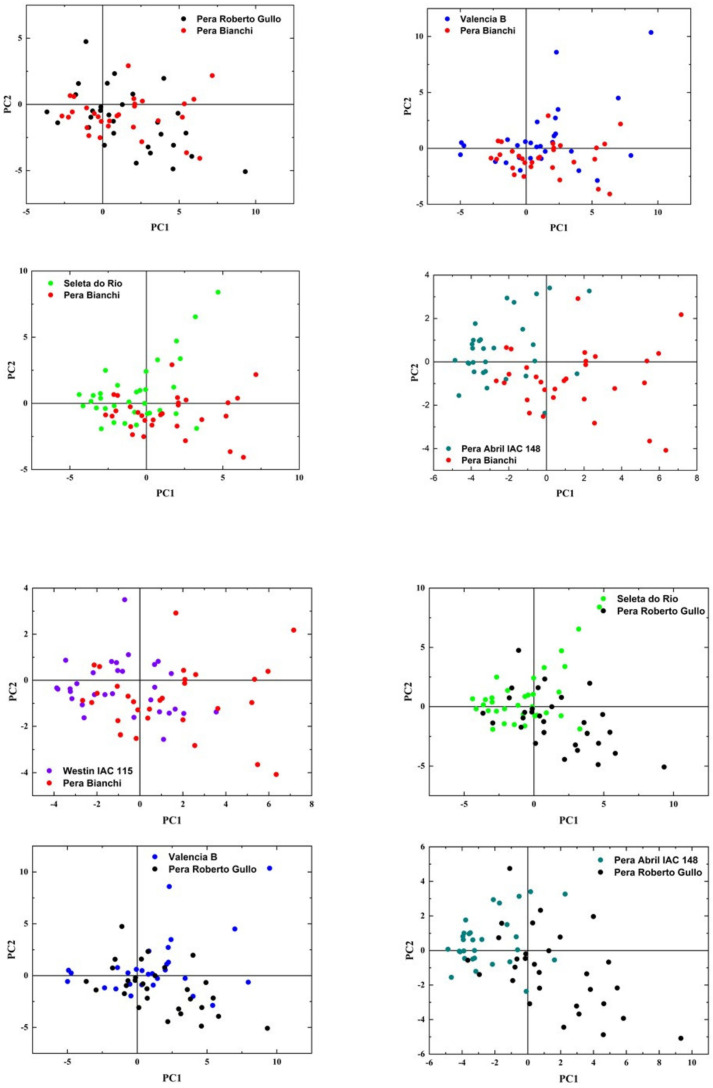
PCA 2-dimensional loading plots of PC2 versus PC1 compared by pair for Valencia B, Pera Abril IAC 148, and Westin IAC 115 accessions of the Common variety of set 2.

**Figure 7 molecules-26-03092-f007:**
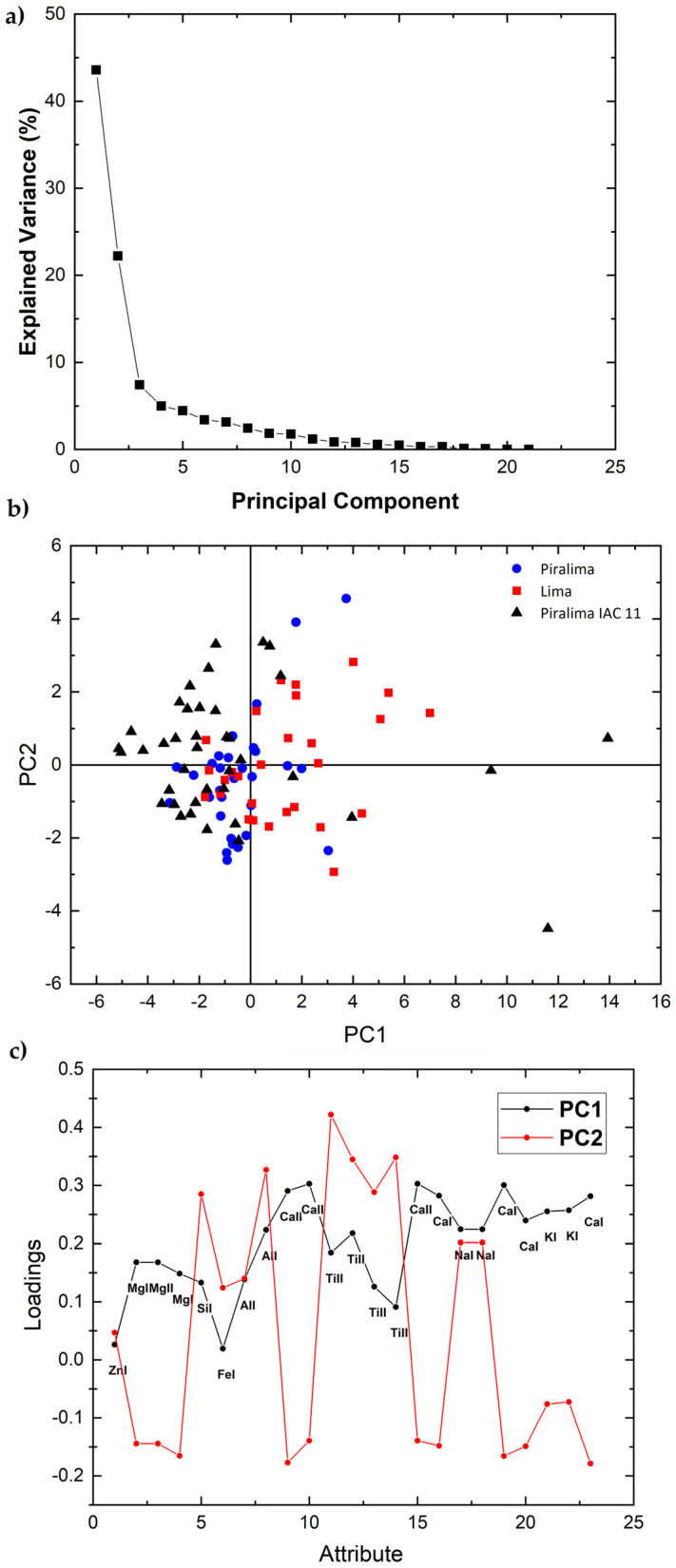
Scree plot (**a**), PCA 2-dimensional score plot of PC2 versus PC1 (**b**), and loading plot showing how each variable correlates to each PC (**c**) for the three accessions of the Low Acidity variety of set 2.

**Table 1 molecules-26-03092-t001:** The twenty-three emission lines identified in the LIBS spectrum of a sweet orange leaf, which were relevant for the discrimination among the accessions of sweet orange. The roman number I after the element symbol indicates an atomic line and II an ionic line. LIBS emission line wavelength corresponds to the maximum intensity of that line.

Element	LIBS Emission Wavelength (nm)	Element	LIBS Emission Wavelength (nm)
Zn I	202.55	Ti II	337.27
Mg II	279.59	Ti II	338.37
Mg II	280.34	Ca II	422.65
Mg I	285.31	Ca I	585.70
Si I	288.24	Na I	588.98
Fe I	302.10	Na I	589.57
Al I	308.23	Ca I	643.83
Al I	309.24	Ca I	646.19
Ca II	315.90	K I	766.53
Ca II	317.94	K I	769.95
Ti II	334.93	Ca I	854.28
Ti II	336.13		

**Table 2 molecules-26-03092-t002:** Correctly classified instances obtained by constructing an induced classifier based on the calculated peak areas as input variables.

Set 1		Set 2	
Leaf Samples	Correctly Classified Instances	Leaf Samples	Correctly Classified Instances
Common	93%	Common	65%
Navel	100%	Low Acidity	86.5%
Pigmented	90%		

**Table 3 molecules-26-03092-t003:** Confusion matrix achieved by the classification model applied to all accessions of set 1 evaluated together. All values are expressed in percentage (%). Samples are indicated as follows. Navel variety: A, Baianinha Piracicaba; B, Baia Vale del Cauca. Common variety: C, Pera Ipigua; D, Ovale; E, Pera Mahle; F, Pera Olimpia. Pigmented variety: G, Sanguinea; H, Red Pulp of Mombuca; I, Sanguinelli; J, Doppio Red Pulp.

	A	B	C	D	E	F	G	H	I	J
A	100	0	0	0	0	0	0	0	0	0
B	0	100	0	0	0	0	0	0	0	0
C	0	0	100	0	0	0	0	0	0	0
D	0	0	0	100	0	0	0	0	0	0
E	0	0	0	0	100	0	0	0	0	0
F	0	0	0	0	0	82.6	13.4	0	0	4.4
G	0	0	0	0	0	16.7	66.6	12.5	0	4.2
H	0	0	0	0	0	4.3	0	91.4	0	0
I	0	0	0	0	0	0	0	0	100	0
J	0	0	0	0	0	0	0	0	0	100

**Table 4 molecules-26-03092-t004:** Confusion matrix achieved by the classification model applied to all accessions of set 2 evaluated together. All values are expressed in percentage (%). Samples are indicated as follows. Low Acidity variety: A, Piralima IAC 2; B, Lima IAC 9; C, Piralima IAC 11. Common variety: D, Pera Abril IAC 148; E, Valência B; F, Westin IAC 115; G, Seleta do Rio; H, Pera Roberto Gullo; I, Pera Bianchi. Pigmented variety: J, Valência Puka.

	A	B	C	D	E	F	G	H	I	J
A	83.3	0	6.7	6.7	3.3	0	0	0	0	0
B	3.6	85.7	0	3.6	7.1	0	0	0	0	0
C	0	0	79.5	0	0	0	2.6	7.7	0	10.3
D	0	0	3.3	80	0	0	18.2	6.7	0	0
E	0	3.4	6.9	0	75.9	6.9	0	0	6.9	0
F	12.9	3.2	0	0	3.2	77.5	0	0	0	3.2
G	0	0	21.2	18.2	0	0	54.5	3	0	3
H	0	0	0	0	0	0	6.7	90	0	3.3
I	0	0	0	0	0	0	0	0	100	0
J	0	0	3.3	3.7	0	0	3.3	3.3	0	86.7

**Table 5 molecules-26-03092-t005:** Confusion matrix achieved for the Common variety accessions of set 1: A, Pera Ipigua, B, Ovale, C, Pera Mahle and D, Pera Olímpia.

	A	B	C	D
A	100%	0	0	0
B	0	100%	0	0
C	0	0	100%	0
D	0	0	0	100%

**Table 6 molecules-26-03092-t006:** Confusion matrix achieved for the Common variety accessions of set 2: A, Pera Abril IAC 148, B, Valencia B, C, Westin IAC 115, D, Seleta do Rio, E, Pera Roberto Gullo and F, Pera Bianchi.

	A	B	C	D	E	F
A	70%	5%	0	20%	5%	0
B	5%	85%	5%	0	0	5%
C	0	0	90%	0	0	5%
D	20%	0	0	70%	5%	5%
E	0	0	0	15%	85%	0
F	0	5%	0	0	0	95%

**Table 7 molecules-26-03092-t007:** Confusion matrix achieved for the Pigmented variety accessions of set 1: A, Sanguinea, B, Red Pulp of Mombuca, C, Sanguinelli and D, Doppio Red Pulp.

	A	B	C	D
A	100%	0	0	0
B	8.7%	91.3%	0	0
C	0	0	100%	0
D	0	0	0	100%

**Table 8 molecules-26-03092-t008:** Confusion matrix achieved for the Navel variety accessions of set 1: A, Baianinha Piracicaba, B, Bahia Vale del Cauca.

	A	B
A	100%	0
B	0	100%

**Table 9 molecules-26-03092-t009:** Confusion matrix achieved for the Low Acidity variety accessions of set 2: A, Piralima IAC 2, B, Lima IAC 9 and C, Piralima IAC 11.

	A	B	C
A	90%	10%	0
B	20%	80%	0
C	0	0	100%

**Table 10 molecules-26-03092-t010:** Set 1 of sweet orange accessions of the Common, Pigmented, and Navel varieties and set 2 of sweet orange accessions of the Common, Pigmented, and Low Acidity varieties analyzed by LIBS.

Variety	Accessions
	**Set 1**
Common	Pera Ipigua, Pera Mahle, Ovale, Pera Olímpia
Pigmented	Sanguinea, Red Pulp of Mombuca, Sanguinelli, Doppio Red Pulp
Navel	Baianinha Piracicaba, Baia Vale del Cauca
	**Set 2**
Common	Pera Abril IAC 148, Pera Bianchi, Valência B, Westin IAC 115, Pera Roberto Gullo, Seleta do Rio
Pigmented	Valência Puka
Low Acidity	Lima IAC 9, Piralima IAC 11, Piralima IAC 2

## Data Availability

Data is contained within the article.

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
