# Peer review of "Discrimination of Genetically Very Close Accessions of Sweet Orange (Citrus sinensis L. Osbeck) by Laser-Induced Breakdown Spectroscopy (LIBS)"

_molecules, 2021, doi:10.3390/molecules26113092_

Round 1
Reviewer 1 Report
I have reviewed the manuscript “Discrimination of genetically very close accession of sweet orange (Citrus sinensis L. Osbeck) by laser-induced breakdown spectroscopy” by Magalhães et al. The paper deals with the application of laser-induced plasma spectroscopy (LIBS) in agriculture. The research is mainly based on the direct analysis of plant leaves for the discrimination of various accessions of sweet oranges at the nursery stage by using LIBS spectral fingerprints together with chemometric methods.
Despite some investigations have already been reported in the pertinent literature, I consider that further research on this topic is still useful to provide a deeper insight into the improvement of LIBS applications aimed at transportable instruments for analysis in situ.
Based on these considerations, I suggested the editor that the article should be accepted for publication with the suggestion of some minor changes and clarifications.
- Introduction:
The following reviews devoted to the state-of-the-art of LIBS technique in the field of agriculture should be cited.
- D. Santos Jr. et al., Laser-induced breakdown spectroscopy for analysis of plant materials: A review, Spectro chimica Acta Part B 71–72 (2012) 3–13.
Peng, Challenging application for multi-element analysis by laser-induced breakdown spectroscopy in agriculture: A review, Trends in Analytical Chemistry 85 (2016) 260–272.
- Please, also highlight briefly how this new study contributes to the advancement of this field.
- Results and discussion
P.2, lines 83, 84: The spectral ranges of the seven spectrometers should be moved to the Materials and Methods section.
Fig. 1: The main spectral lines detected (listed in Table 1) should be properly marked in the example spectrum of the orange leaf. There were differences with the other spectra measured? Please, also clarify this point in the manuscript.
Table 1: The elements detected in leaves are macronutrients, micronutrients, or toxic elements? Please, clarify this point in the manuscript.
It is possible that the presence of some toxic elements, if a given limit is surpassed, affects the accession growing?
Figs. 2 to 10: The clusters should be drawn in the graphs for improving clarity.
Fig. 6 to 8: The fonts of the legends are fuzzy and should be improved.
P. 10, line 158: Please, replace “areas” with “peak areas”.
P. 11, line 172: “… each combination of rootstock and canopy.” This phrase is repeated many times throughout the text. Please write it once and delete the others.
- Materials and methods:
- 13, line 233: This section should be numbered with 3.
Why the leave samples analyzed were divided in two sets for winter and summer? Please, explain in the text.
P. 13, line 248: Table 1 should be corrected by Table 10 in the table caption.
P. 14, line 251: Table 2 should be corrected by Table 11 in the table caption.
- Maybe, the two tables can be unified in a single table?
The leaves were analyzed in the form of fresh, dried, powder, pellet, other? Why?
Was the spectrometer time-gated or had an initial delay time to avoid the bremsstrahlung radiation? Please include this data in the text.
P. 14, line 272: “…emitting an electromagnetic radiation”. Please, delete “an”.
P. 14, line 273: “… spectrometer that resolves the atomic emissions.” Please add “in wavelength” at the end of the sentence.
P. 14, line 275: “… the intensity of which is proportional to the element concentration”.
This is valid if self-absorption and matrix effects are negligible. Was the case of the experiment?
The multivariate analyses can be influenced by matrix effects (an important concern for analysis of vegetables) or self-absorption? Please, clarify this point in the text.
P. 15, line 292: In spectra recorded with low resolution instruments the instrumental broadening dominates and so the experimental line profiles are approximated by a Gaussian function. Why the line areas were obtained by fitting with a Lorentzian function?
- Conclusions:
P.16, line 340: This section should be numbered 4.
Author Response
Reviewer 1
I have reviewed the manuscript “Discrimination of genetically very close accession of sweet orange (Citrus sinensis L. Osbeck) by laser-induced breakdown spectroscopy” by Magalhães et al. The paper deals with the application of laser-induced plasma spectroscopy (LIBS) in agriculture. The research is mainly based on the direct analysis of plant leaves for the discrimination of various accessions of sweet oranges at the nursery stage by using LIBS spectral fingerprints together with chemometric methods.
Despite some investigations have already been reported in the pertinent literature, I consider that further research on this topic is still useful to provide a deeper insight into the improvement of LIBS applications aimed at transportable instruments for analysis in situ.
Based on these considerations, I suggested the editor that the article should be accepted for publication with the suggestion of some minor changes and clarifications.
Introduction:
The following reviews devoted to the state-of-the-art of LIBS technique in the field of agriculture should be cited.
- Santos Jr. et al., Laser-induced breakdown spectroscopy for analysis of plant materials: A review, Spectro chimica Acta Part B 71–72 (2012) 3–13.
Peng, Challenging application for multi-element analysis by laser-induced breakdown spectroscopy in agriculture: A review, Trends in Analytical Chemistry 85 (2016) 260–272.
Please, also highlight briefly how this new study contributes to the advancement of this field.
Response 1: As requested, a paragraph has been added in the Introduction of the revised text concerning the state of the art of LIBS applied to agriculture and citing the two reviews suggested.
Results and discussion
P.2, lines 83, 84: The spectral ranges of the seven spectrometers should be moved to the Materials and Methods section.
Response 2: The spectral ranges have been moved to Materials and Methods section as suggested by the reviewer.
Fig. 1: The main spectral lines detected (listed in Table 1) should be properly marked in the example spectrum of the orange leaf. There were differences with the other spectra measured? Please, also clarify this point in the manuscript.
Response 3: As suggested, Figure 1 has been replaced by two expanded Figures that visualize better the LIBS spectrum, in which the relevant emission lines of interest are identified.
Table 1: The elements detected in leaves are macronutrients, micronutrients, or toxic elements? Please, clarify this point in the manuscript.
Response 4: As suggested, the detected elements were classified as macro, micro and toxic elements in the revised text.
It is possible that the presence of some toxic elements, if a given limit is surpassed, affects the accession growing?
Response 5: In any case, if the amount of an element is over a certain limit, it results toxic to the plant. However, the amount of all elements in all accessions studied were within the standard nutritional limits.
Fig. 6 to 8: The fonts of the legends are fuzzy and should be improved.
Response 6: All fonts have been improved in the revised manuscritpt.
- 10, line 158: Please, replace “areas” with “peak areas”.
Response 7: “Areas” have been replaced by “peak areas”.
- 11, line 172: “… each combination of rootstock and canopy.” This phrase is repeated many times throughout the text. Please write it once and delete the others.
Response 8: Corrected.
Materials and methods:
- 13, line 233: This section should be numbered with 3.
Response 9: Corrected.
Why the leave samples analyzed were divided in two sets for winter and summer? Please, explain in the text.
Response 10: The twenty accessions were divided into two sets, i.e. summer and winter, according to the period of year in which leaves were collected. This was done to test the classifiers in two different situations, as the season, i.e. the amount of rain, temperature, humidity, availability of nutrients, influences the spectral characteristics of the samples.
- 13, line 248: Table 1 should be corrected by Table 10 in the table caption.
Response 11: Corrected.
- 14, line 251: Table 2 should be corrected by Table 11 in the table caption. Maybe, the two tables can be unified in a single table?
Response 12: As suggested, the two Tables have been merged in only one new Table 10.
The leaves were analyzed in the form of fresh, dried, powder, pellet, other? Why?
Response 13: All measurements were performed on fresh leaves aiming at minimal sample preparation in order to test the possibility to use the LIBS technique directly in the field.
Was the spectrometer time-gated or had an initial delay time to avoid the bremsstrahlung radiation? Please include this data in the text.
Response 14: The spectrometer used in this study was a LIBS2500 from Ocean Optics, which is not time-gated, whereas the delay time was settled at 2.5 µs. These details have been included in the revised text.
- 14, line 272: “…emitting an electromagnetic radiation”. Please, delete “an”.
Response 15: Done.
- 14, line 273: “… spectrometer that resolves the atomic emissions.” Please add “in wavelength” at the end of the sentence.
Response 16: Done.
- 14, line 275: “… the intensity of which is proportional to the element concentration”.
This is valid if self-absorption and matrix effects are negligible. Was the case of the experiment? The multivariate analyses can be influenced by matrix effects (an important concern for analysis of vegetables) or self-absorption? Please, clarify this point in the text.
Response 17: As we worked with selected areas, self-absorption and matrix effects were negligible or greatly minimized. Furthermore, the leaf samples were collected from plant grown in a greenhouse, thus genetically very close so matrix differences should not be too large. As it regards self-absorption effects, the correlation at the point decreases thus it was not possible to have sufficient sensibility to detect any concentration variation.
- 15, line 292: In spectra recorded with low resolution instruments the instrumental broadening dominates and so the experimental line profiles are approximated by a Gaussian function. Why the line areas were obtained by fitting with a Lorentzian function?
Response 18: The peak areas fit was evaluated by the MATLAB function “peak fit” testing several functions including the Gaussian function. For each function the error was evaluated and the best adjusted function in the case of our spectrometer resulted to be the Lorentzian function.
A paragraph on these aspects was added in the revised text in section 3.3.
- Conclusions:
P.16, line 340: This section should be numbered 4.
Response 19: Done.
Reviewer 2 Report
In this work LIBS is employed to classify of accessions of sweet orange varieties. This manuscript presents an interesting work and should be published in Molecules after some corrections.
In the introduction section, a bibliographic research of using LIBS for plant analysis should be included and the different multivariate statistical techniques employed to classify samples.
In section 4.2 add the focusing lens distance. What is the distance from the focusing lens to the sample? Clarify how the laser fluence was obtained.
Table 1 only present the analyzed transitions but not all the observed ones (In Fig. 1 other transitions could also be observed). Please clarify this point in the text.
The description of PCA method is missing. Please describe which kind of matrix was used for the statistical processing (variables, cases, correlation or covariance). Were the used results subdued to autoscaling before insertion in matrix?
Detailed description of PCA results is missing. Please insert the loading factors near the PCA scores, add the total variance and explain it in more detail. What are the main elements observed that allow the identification of the different samples? For example, Mg, Ti, Na, etc. Besides, I suggest that you modify the abstract and conclusions according to the obtained results.
Author Response
Dear Reviewers,
Thank you so much for your review efforts and constructive comments about our manuscript. The suggestions were very helpful in improving its quality.
Please find below our responses to your comments.
Yours sincerely,
The Authors
Reviewer 2
Comments and Suggestions for Authors
In this work LIBS is employed to classify of accessions of sweet orange varieties. This manuscript presents an interesting work and should be published in Molecules after some corrections.
- In the introduction section, a bibliographic research of using LIBS for plant analysis should be included and the different multivariate statistical techniques employed to classify samples.
Response 1: As requested, a paragraph has been added in the Introduction of the revised text concerning the state of the art of LIBS applied to plants.
- In section 4.2 add the focusing lens distance. What is the distance from the focusing lens to the sample? Clarify how the laser fluence was obtained.
Response 2: The focusing lens distance was 75 mm. The laser fluence was calculated by the equation: fluence [J/cm^2] = total energy/effective area of the beam. The beam area was determined by the knife edge method fully described in “Suzaki, Y., Tachibana, A. Measurement of the µm sized radius of Gaussian laser beam using the scanning knife-edge. Applied Optics, v. 14, n.12, p. 2809-2810, 1975”. These details have been included in the revised text.
- Table 1 only present the analyzed transitions but not all the observed ones (In Fig. 1 other transitions could also be observed). Please clarify this point in the text.
Response 3: By purpose, we have preferred to focus only on those transitions of elements relevant to our study, and not consider the others.
- The description of PCA method is missing. Please describe which kind of matrix was used for the statistical processing (variables, cases, correlation or covariance). Were the used results subdued to autoscaling before insertion in matrix?
Response 4: As requested, a new paragraph describing the PCA method has been added in section 3.3.
- Detailed description of PCA results is missing. Please insert the loading factors near the PCA scores, add the total variance and explain it in more detail. What are the main elements observed that allow the identification of the different samples? For example, Mg, Ti, Na, etc. Besides, I suggest that you modify the abstract and conclusions according to the obtained results.
Response 5: The scree plots and loading plots have been added to the PCA score plots in Figures 2 to 5, and 7, and the related text has been revised.
Reviewer 3 Report
Dear Editor,
We report on manuscript “Molecules-1201962”, entitled:
“Discrimination of genetically very close accessions of sweet orange (Citrus sinensis L. Osbeck) by laser-induced breakdown spectroscopy (LIBS)”,
authored by Magalhães A. B., et al., submitted to your journal for consideration.
In this work, LIBS is used to discriminate orange leaves from different orange trees. The analysis of the spectroscopic LIBS data is performed by implementing Principal Component Analysis (PCA) for dimensionality reduction and visual inspection, while a classification scheme, based on Classification via Regression (CVR) and Partial Lest Squares (PLS) is proposed for discriminating the different accessions of orange trees based on the different orange tree varieties. The work is interesting, although the number of samples is rather limited for such a study. The present manuscript can be accepted after some revisions which are stated explicitly below.
Comments regarding this work.
- There is no discussion regarding the physical meaning concerning the discrimination of sweet orange leaves.
- The LIBS spectra are hardly explained and there is not any comment regarding the importance/variance of the spectroscopic features used in the PCA analysis.
- The gating conditions (i.e. integration time and gate width) are note reported or commented. Are they tunable or they are fixed by the manufacturer of the commercial system? If they are tunable, do the authors have optimize their selection? This is important as from the shown spectra in Figure 1, it is apparent the presence of strong continuum which makes difficult and ambiguous the quantitative analysis e.g. of the total intensity or the peak intensity of the spectral lines. Besides, the quality of Fig. 1 is very poor and the spectral peaks are hardly seen. Larger dimensions image and maybe its splitting in two parts could better serve the purposes.
- Are the intensities of the spectral lines of the different spectral windows directly comparable? Are the CCDs of the seven spectrometers identical or different? If not do they have been calibrated by the authors somehow, e.g. by a black body source (e.g. a tungsten lamp) or else?
- The 2D-PCA score plots of all Figures, i.e. from Fig. 2 to Fig. 9, are very confusing and not readable. Please use solid symbols and/or colors, otherwise they cannot be read.
- It seems that no external validation has been used in the classification models. Only k-fold stratified cross-validation seems to have been used. External validation is a must in machine learning applications because cross-validation can lead to overestimating the predictive models’ capabilities in accurately predicting new, previously unseen from the algorithm, spectroscopic data. Please comment and add this important missing procedure.
Some comments regarding the implementation of PCA:
The authors used the calculated areas from 22 peaks, i.e., spectral lines, as inputs to the PCA algorithm, according to:
page: 15, lines: 294-295: “The calculated areas, A1, for each selected element peak from all samples, i.e. 22 peaks for each sample, were subjected to PCA”
Presumably, these peaks correspond to the peaks enlisted in Table 1. However, Table 1, enlists 23 peaks. So, which peaks did the authors explicitly chose? Moreover, Figure 1 shows a LIBS spectrum obtained from a sweet orange leaf. However, for the inexperienced reader such a Figure would be confusing. We suggest the authors add annotations of the identified spectral lines in Figure 1, so that they can be clearly seen. Moreover, please note that there are typos in this Figure (Wavelenght instead of Wavelength and u.a. instead of a.u.).
The authors use PCA to examine if any cluster formation occurs within the LIBS spectroscopic data, based on the different orange accessions. The transformation of the data by PCA is based on the features’ variance. Which features, i.e., spectral lines, have the highest variance? This can be visualized by providing the loadings of each Principal Component.
We highly encourage the authors to use colored score plots. Unfortunately, in many of the plots of the manuscript the data points are difficult to distinguish because either overlap occurs, or the plots have bad quality. Some of the plots, such as the bottom right of Fig. 6, have such a low quality that one can hardly understand them.
Page: 4, lines: 100 - 101. “In all cases the total variance retained in the first two PCs did not increase by increasing the number of input attributes”
What are the explained variances for each one of the Principal Components used in the score plots? We suggest that the authors include the explained variance percentages on each plot.
Page: 4, lines: 103 – 104. “Although the clusters were somehow overlapping, the centers of each cluster were slightly displaced”
This is not quite evident. How do the authors define the center of the cluster? Did they, somehow, computed group centroids? What method did they use?
Page: 6, lines 121-127: “The main results were: (i) the cluster mass center of Pera Bianchi accession showed a tendency to separate from those of Seleta do Rio, Pera Abril IAC 148 and Valencia B accessions; (ii) the Pera Roberto Gullo accession cluster tended to separate from almost all other accessions, except the Pera Bianchini one, which suggested that this accession possessed characteristics different from the other ones; (iii) the Valencia B accession cluster was slightly separated only from Pera April IAC 148, thus it seemed to be very similar to the others.”
(i-iii): We disagree. These PCA plots seem to contain highly overlapped points, as well as, contain some outliers (possibly). Similar comments apply to the rest of the PCA plots.
Some comments regarding the implementation of the classification model(s):
The classification via regression procedure (CVR) is not clearly explained. CVR seems to be a classifier provided by the software the authors used (i.e., Weka software). However, no details regarding its implementation are provided.
What are the sizes of the datasets used for classification?
The confusion matrices, i.e., Tables 3-9, should indicate the actual and predicted classes.
Author Response
Dear Reviewers,
Thank you so much for your review efforts and constructive comments about our manuscript. The suggestions were very helpful in improving its quality.
Please find below our responses to your comments.
Yours sincerely,
The Authors
Reviewer 3
Dear Editor,
We report on manuscript “Molecules-1201962”, entitled:
“Discrimination of genetically very close accessions of sweet orange (Citrus sinensis L. Osbeck) by laser-induced breakdown spectroscopy (LIBS)”,
authored by Magalhães A. B., et al., submitted to your journal for consideration.
In this work, LIBS is used to discriminate orange leaves from different orange trees. The analysis of the spectroscopic LIBS data is performed by implementing Principal Component Analysis (PCA) for dimensionality reduction and visual inspection, while a classification scheme, based on Classification via Regression (CVR) and Partial Lest Squares (PLS) is proposed for discriminating the different accessions of orange trees based on the different orange tree varieties. The work is interesting, although the number of samples is rather limited for such a study. The present manuscript can be accepted after some revisions which are stated explicitly below.
Comments regarding this work.
There is no discussion regarding the physical meaning concerning the discrimination of sweet orange leaves.
Response 1: This aspect has been developed in the Introduction. Sweet orange is the most important variety for orange juice export market. However, the cultivar identification based on morphological, physiological and agronomic characters can be done only 3-4 years after planting, when already in production. Thus, it is desirable that nurserymen have efficient tools for the correct identification of their plants i.e., the development of economically viable analytical strategies is necessary to advance towards an innovative citriculture. In this work, we have successfully tested the LIBS technique as a rapid means to distinguish between sweet orange cultivars which is expected to be useful for their certification.
The LIBS spectra are hardly explained and there is not any comment regarding the importance/variance of the spectroscopic features used in the PCA analysis.
Response 2: As suggested, further details of LIBS spectra and spectroscopic features used in PCA have been added in the revised text.
The gating conditions (i.e. integration time and gate width) are note reported or commented. Are they tunable or they are fixed by the manufacturer of the commercial system? If they are tunable, do the authors have optimize their selection? This is important as from the shown spectra in Figure 1, it is apparent the presence of strong continuum which makes difficult and ambiguous the quantitative analysis e.g. of the total intensity or the peak intensity of the spectral lines. Besides, the quality of Fig. 1 is very poor and the spectral peaks are hardly seen. Larger dimensions image and maybe its splitting in two parts could better serve the purposes.
Response 3: The gating conditions (integration time, 1 sec) were not tunable in the system used, except the delay time that could be controlled by the laser Q-Switch and was settled at 2.5 µs. The correction of the peaks was done locally in order to avoid the background emission effect on the calculations of peak areas. As suggested, a better Figure 1 with the LIBS spectrum splitted in two enlarged parts with indication of the relevant emission lines of interest has been provided in the revised manuscript.
Are the intensities of the spectral lines of the different spectral windows directly comparable? Are the CCDs of the seven spectrometers identical or different? If not do they have been calibrated by the authors somehow, e.g. by a black body source (e.g. a tungsten lamp) or else?
Response 4: The system used was composed of seven separated CCD spectrometers, thus the intensities of the spectral lines were not directly comparable. For this reason, a normalization was performed separately for each spectrometer range, and the peak areas were properly selected and individually calculated. The Ocean Optics spectrometers used in this work were not calibrated, however they do not lose their calibration in wavelength.
The 2D-PCA score plots of all Figures, i.e. from Fig. 2 to Fig. 9, are very confusing and not readable. Please use solid symbols and/or colors, otherwise they cannot be read.
Response 5: Better Figures have been provided in the revised manuscript.
It seems that no external validation has been used in the classification models. Only k-fold stratified cross-validation seems to have been used. External validation is a must in machine learning applications because cross-validation can lead to overestimating the predictive models’ capabilities in accurately predicting new, previously unseen from the algorithm, spectroscopic data. Please comment and add this important missing procedure.
Response 6: The authors agree with the Reviewer on the importance of an external validation in machine learning applications but in our case the number of samples was small, i.e. about 10 leaves for each accession, thus the authors used the stratified cross-validation as a preliminary attempt to validate the LIBS technique.
Some comments regarding the implementation of PCA:
The authors used the calculated areas from 22 peaks, i.e., spectral lines, as inputs to the PCA algorithm, according to:
page: 15, lines: 294-295: “The calculated areas, A1, for each selected element peak from all samples, i.e. 22 peaks for each sample, were subjected to PCA”
Presumably, these peaks correspond to the peaks enlisted in Table 1. However, Table 1, enlists 23 peaks. So, which peaks did the authors explicitly chose? Moreover, Figure 1 shows a LIBS spectrum obtained from a sweet orange leaf. However, for the inexperienced reader such a Figure would be confusing. We suggest the authors add annotations of the identified spectral lines in Figure 1, so that they can be clearly seen. Moreover, please note that there are typos in this Figure (Wavelenght instead of Wavelength and u.a. instead of a.u.).
Response 7: Sorry, the correct number of peaks is 23, as corrected in the revised text.
Response 8: As suggested, a better Figure 1 with the LIBS spectrum splitted in two enlarged parts with indication of the relevant emission lines of interest has been provided in the revised manuscript. Typos have been corrected.
The authors use PCA to examine if any cluster formation occurs within the LIBS spectroscopic data, based on the different orange accessions. The transformation of the data by PCA is based on the features’ variance. Which features, i.e., spectral lines, have the highest variance? This can be visualized by providing the loadings of each Principal Component.
Response 9: The scree plots and loading plots have been added to the PCA score plots in Figures from 2 to 5, and 7. The related text has been revised with the features variance explained.
We highly encourage the authors to use colored score plots. Unfortunately, in many of the plots of the manuscript the data points are difficult to distinguish because either overlap occurs, or the plots have bad quality. Some of the plots, such as the bottom right of Fig. 6, have such a low quality that one can hardly understand them.
Response 10: Figures have been improved at our best using colors.
Page: 4, lines: 100 - 101. “In all cases the total variance retained in the first two PCs did not increase by increasing the number of input attributes”
What are the explained variances for each one of the Principal Components used in the score plots? We suggest that the authors include the explained variance percentages on each plot.
Response 11: The relevant scree plots have been added in Figures 2 to 5 and 7, and reported in the revised text.
Page: 4, lines: 103 – 104. “Although the clusters were somehow overlapping, the centers of each cluster were slightly displaced”
This is not quite evident. How do the authors define the center of the cluster? Did they, somehow, computed group centroids? What method did they use?
Page: 6, lines 121-127: “The main results were: (i) the cluster mass center of Pera Bianchi accession showed a tendency to separate from those of Seleta do Rio, Pera Abril IAC 148 and Valencia B accessions; (ii) the Pera Roberto Gullo accession cluster tended to separate from almost all other accessions, except the Pera Bianchini one, which suggested that this accession possessed characteristics different from the other ones; (iii) the Valencia B accession cluster was slightly separated only from Pera April IAC 148, thus it seemed to be very similar to the others.”
(i-iii): We disagree. These PCA plots seem to contain highly overlapped points, as well as, contain some outliers (possibly). Similar comments apply to the rest of the PCA plots.
Response 12: Although we substantially agree with the Reviewer comments, results of PCA show a preliminary tendency of separating these groups. The overlapping and possible presence of the outliers may be due to the large number of data and correlated variables. Thus, to improve the discrimination among the sweet orange accessions of the two sets, a classification model combining CVR and PLSR was employed as machine learning can deal better than PCA with a large number of highly correlated variables. The results achieved, even if considered only approximate, are of some indicative practical value as the morphological features of accessions are very close and discrimination is not possible by using DNA markers.
Some comments regarding the implementation of the classification model(s):
The classification via regression procedure (CVR) is not clearly explained. CVR seems to be a classifier provided by the software the authors used (i.e., Weka software). However, no details regarding its implementation are provided.
Response 13: CVR was based on the combination of two approaches: i.e. a model tree algorithm and a regression function via PLSR. The input data were exported to an open source software Weka for the induction of the classifier, which was based on the elemental concentration variation in the different accessions, and allowed to obtain the confusion matrices. These details are provided in the revised manuscript.
What are the sizes of the datasets used for classification?
Response 14: Two datasets were used in the two attempts: in the first one 23 single peak areas, and in the second one the whole spectrum consisting of 13700 variables.
The confusion matrices, i.e., Tables 3-9, should indicate the actual and predicted classes.
Response 15: The matrix rows indicate the percent of leaf samples belonging to the actual class, the matrix columns indicate the percentage of samples predicted by the model, and the values on the matrix diagonal indicate the correctness of the model. The corresponding text has been improved with relevant details.
Round 2
Reviewer 1 Report
I checked the revised manuscript and I consider that it is improved and should be published in the present form.